# Bonding and Thermal-Mechanical Property of Gradient NiCoCrAlY/YSZ Thermal Barrier Coatings with Millimeter Level Thickness

Yu Wang, Qi Liu, Quansheng Zheng, Tianqing Li, Nanjing Chong and Yu Bai *

State Key Laboratory for Mechanical Behavior of Materials, Xi'an Jiaotong University, Xi'an 710049, China; wangyu0730@mail.xjtu.edu.cn (Y.W.); lq925742369@stu.xjtu.edu.cn (Q.L.); zqs321@stu.xjtu.edu.cn (Q.Z.); litianqing@stu.xjtu.edu.cn (T.L.); njchong@stu.xjtu.edu.cn (N.C.)
* Correspondence: byxjtu@mail.xjtu.edu.cn

**Abstract:** The thermal insulation properties of thermal barrier coatings (TBCs) can be significantly improved with increasing the coating thickness. However, due to the weak bonding of high-thickness TBCs, the low reliability and short lifetime greatly limits their application under some severe operating conditions. In this study, a novel and high-efficiency synchronous dual powder feeding method is used to deposit a series of gradient NiCoCrAlY/YSZ coatings with millimeter level thickness. The tensile bonding strengths and residual stress state of coatings are evaluated in order to explore the effect of thickness on the bonding strength of coatings. The results suggested that, due to some micro-convex structure at the "GC/TC" interface and inside "GC" layer, the bonding strength of 1000-μm-thickness gradient NiCoCrAlY/YSZ TBCs with the 4:6 and 2:8 hybrid ratios is over 44 MPa compared to the common TBCs. The fracture position gradually shifts from NiCoCrAlY bond coat to NiCoCrAlY/YSZ transition zone and finally to the YSZ top coat owing to the different position of residual stress concentrations. After thermal cycling tests, the 1000-μm-thickness gradient coating exhibits a higher thermal cycling life. Some coarse cracks initiate and propagate at the bottom region of TBCs, which is mainly due to thermal expansion mismatch stress that finally results in the failure of the gradient coating between the "BC" layer and the substrate.

**Keywords:** gradient coating; high thickness; synchronous feeding; plasma spraying; bonding strength; thermal-mechanical property

## 1. Introduction

Thermal barrier coatings (TBCs) are widely used on aircraft and the gas turbine blades to protect metallic components against the high-temperature environment in which they work [1–4]. The thickness of coating is a key factor influencing the performance of TBCs. Previous studies have indicated that for every 0.1 mm thickness one can obtain an approximately 30 °C drop in the average temperature of the substrate [5]. However, due to the accumulation of thermal residual stress during plasma spraying, it is very difficult to deposit high-thickness TBCs with the excellent bonding strength and thermal-mechanical properties. The research by Peng et al. found that the maximum bonding strength of 240-μm-thickness coating was approximately 41 MPa and it rapidly decreased to 25 MPa when the coating thickness increased to 750 μm [6]. The weak bonding strength for the thickness coatings was associated with the high residual stress due to the coating-substrate thermal expansion mismatch and the fast formation of flattened droplets [6]. However, for the high-thickness coatings, Guo et al.'s research found that some vertical cracks within the coating could release of residual stress and improve the thermal-mechanical properties [7–9] and the increase of substrate temperature further promoted the increase of vertical cracks content [10]. Therefore, the coating microstructure highly influenced the thermal-mechanical properties.

In order to decrease the residual stress, some gradient TBCs were designed. For example, Qin et al. found that the bonding strength of 50 wt.% NiCoCrA1Y/50 wt.% YSZ gradient TBCs could reach a maximum value of approximately 25 MPa and gradually decreased when the coating thickness increased from 1000 to 2250 μm [11]. Besides, some modified methods were applied to improve the bonding properties, such as surface sealing, laser glazing, sol-gel impregnation, etc. [12,13]. Therefore, it is urgent to explore preparation methods for high bonding strength coatings with millimeter-level thickness.

At present, the gradient TBCs are usually fabricated by premixing the starting powders in a certain proportion and preparing a precursor solution [14–16]. In this method, it very difficult to control the spray parameters to form a uniform coating as the two types of starting powders usually have different densities, sizes, morphologies, melting points and flowability. Therefore, new methods need to be explored in order to tailor higher-thickness gradient coatings and simultaneously enhance their bonding strength. Our previous study suggested that the supersonic atmospheric plasma spraying (SAPS) technique can successfully deposit 300-μm-thick $La_2Ce_2O_7$ (LC)/YSZ gradient coatings with the high-performance using a synchronous dual powder system [17]. However, how to fabricate gradient TBCs with millimeter level thickness by this method is still a huge challenge. Besides, the thermal conductivity and thermal cyclic life of high-thickness gradient coatings at elevated temperature have not been evaluated.

Against the above background, in the present work, millimeter-level thickness gradient NiCoCrAlY/YSZ TBCs are fabricated by the SAPS technique. The bonding strength and residual stress state of coatings are systematically evaluated with an aim of illuminating the effect of thickness on the bonding strength and failure mode of the coatings under tensile conditions. The thermal-mechanical properties, including the thermal conductivity and thermal cycling performances also were studied.

## 2. Experimental Procedure

The gradient NiCoCrAlY/YSZ coatings are deposited on nickel-based superalloy GH4169 substrates with 25.4 mm in diameter and 6 mm in thickness.

A commercial NiCoCrAlY powder (4454, Sulzer Metco Inc., Shanghai, China) (see Figure 1a) and an 8 wt.% yttria-stabilized zirconia (Sang Yao Technical Co., Ltd., Beijing, China) powder (see Figure 1b) are used to deposit the bond coat and top coat, respectively.

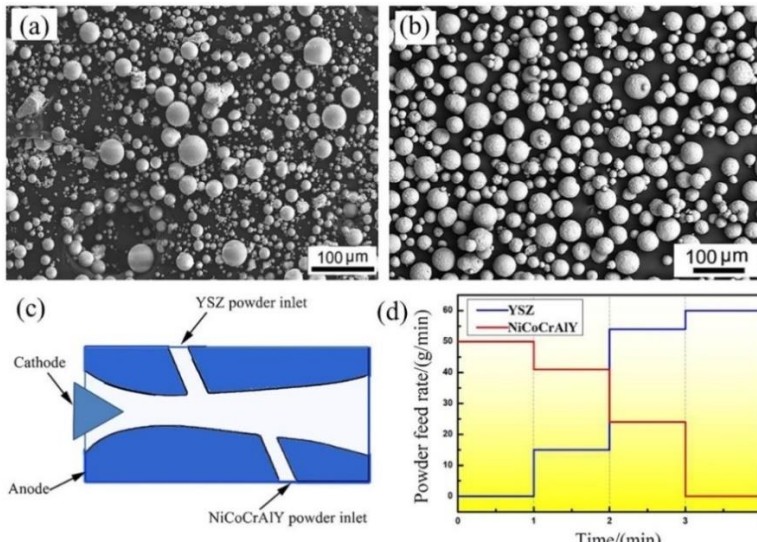

**Figure 1.** Original (**a**) NiCoCrAlY and (**b**) YSZ feedstock powder; (**c**) synchronous dual powder inlets and (**d**) powder feeding rate as a function of spraying time.

The morphologies and high-magnification images of original NiCoCrAlY and YSZ powders are shown in Figure 1a,b, in which the average size is 65 and 45 μm, respectively.

The YSZ feedstock exhibits a near-perfect spherical morphology and the grain size in the feedstock is 150–580 nm.

A high-efficiency supersonic atmospheric plasma spraying (SAPS) system (CYS-120, Sulzer Metco Inc, Beijing, China) equipped with a synchronous dual powder feed system is used to deposit the gradient NiCoCrAlY/YSZ coatings. Two powder inlets which are independently connected to two powder feeders (see Figure 1c) are employed to deposit the gradient TBCs by adjusting the NiCoCrAlY and YSZ feeding rates in a proper ratio. Two powders with different melting points can be simultaneously injected into the different parts of the plasma jet. The powder feeding rate as a function of spray time is shown in Figure 1d. The detailed spray parameters are as follows: primary gas Ar is 130 sLpm, secondary gas $H_2$ is 18 sLpm, the current is 430 A, the voltage is 130 V and the spray distance is 100 mm. The different gradient structures of samples with millimeter level thickness are listed in Table 1. The thickness of TBCs ranges from 1000 μm to 3000 μm. The ratio of gradient layers for NiCoCrAlY and YSZ component is regulated by 4:6, 6:4 and 2:8, respectively.

**Table 1.** The different gradient structures of samples with millimeter level thickness.

| Sample | Thickness/mm | Bond Layer NiCoCrAlY/mm | Gradient Layer (NiCoCrAlY:YSZ)/mm | Top Layer YSZ (mm) |
|--------|--------------|-------------------------|-----------------------------------|--------------------|
| G1 | 1.0 | 0.2 | Layer 1-0.2 (4:6) + Layer 2-0.3 (2:8) | 0.3 |
| G2 | 1.5 | 0.3 | Layer 1-0.3 (4:6) + Layer 2-0.4 (2:8) | 0.5 |
| G3 | 2.0 | 0.2 | Layer 1-0.6 (4:6) + Layer 2-0.4 (2:8) | 0.9 |
| G4 | 2.5 | 0.4 | Layer 1-0.5 (4:6) + Layer 2-0.5 (2:8) | 1.1 |
| G5 | 3.0 | 0.2 | Layer 1-0.8 (4:6) + Layer 2-0.8 (2:8) | 1.2 |
| G6 | 1.5 | 0.3 | Layer 1-0.3 (6:4) + Layer 2-0.3 (4:6) | 0.6 |
| G7 | 2.0 | 0.4 | Layer 1-0.4 (6:4) + Layer 2-0.4 (4:6) | 0.8 |
| G8 | 2.5 | 0.5 | Layer 1-0.6 (6:4) + Layer 2-0.6 (4:6) | 0.8 |

The bonding strength of as-sprayed coatings is measured at room temperature by a tensile tester (model 1195, Instron, Shanghai, China) according to the ASTM 633-79 standard. The samples are joined with cylindrical counterparts using an adhesive film (FM1000, Beijing Qinhe Technology, Co., Ltd., Beijing, China) and followed by heat treatment at 197 °C for 4 h in a furnace to reach the self-adhesive point of the glue. The constant displacement rate is 0.2 mm·min$^{-1}$ and the final results are the average values of three samples. The in-plane residual stress (parallel to the interface) within the top coat is measured by X-ray diffraction (the $\sin^2 \varphi$ method) for as-sprayed and thermal cycled coatings.

The thermal conductivity of coatings is usually related to the thermal diffusivities and density. The thermal diffusivities ($\alpha$) and the specific heat capacity ($C_p$) of samples are measured using a laser flash apparatus (LFA 427, Netzsch, Searle, Germany) at the temperature of 25 °C and 260 °C. Each specimen is measured four times at the corresponding temperature. The density ($\rho$) of samples is measured through the Archimedes method. Thus the thermal conductivity ($\lambda$) is calculated by the following equation [18]:

$$\lambda = \rho \times C_p \times \alpha \tag{1}$$

Thermal shock tests were conducted in a muffle furnace. When the temperature inside the furnace reached 1050 ± 20 °C, the samples are placed in the furnace and held for 5 min. During the cooling stage, the temperature of samples is rapidly cooled down to 25 °C via heat transfer with the surrounding environment. More than 5% of the spallation region of surface of top coating is adopted as the criteria for the failure of the samples.

The microstructure of as-sprayed and failed coatings after tensile test is observed using scanning electron microscopy (SEM, VEGAII XMU, Tescan, Brno, Czech Republic) with an energy dispersive spectrometer (EDS, INCA-AE350, British Oxford Inc., Oxford, UK). Before spraying and failed coatings sectioning, the two-part thermosetting resin is used to protect the microstructure of coatings. Then the resin is treated at 120 °C for 1 h and cross-

sections samples are produced by wet diamond sawing from 600-grit to 1500-grit, followed by polishing and cleaning with ethanol. Ten SEM micrograph images are randomly taken from the cross-section of each as-sprayed gradient coating. The porosity content of coatings is quantitatively calculated by an image analysis (IA) method using Image Pro Plus 6.0 software (Media Cybernetics Inc., Rockville, MD, USA). The porosity of the coatings is also measured by an Archimedes drainage method. A free-standing coating is separated by submerging the sample in 40% hydrochloric acid, and then cleaned and dried in the oven. Before measurement, the samples are boiled in distilled water for 2 h to facilitate pore penetration. A detailed description of the Archimedean technique used to determine the coating porosity is available in the literature [19].

## 3. Results and Discussion

### 3.1. Microstructure of As-Sprayed Gradient Coatings

Figure 2 shows cross-sectional SEM images of gradient NiCoCrAlY/YSZ TBCs with different thicknesses. It can be seen from Figure 2 that the NiCoCrAlY and YSZ phase can be apparently distinguished by their dark or light color. The TBCs can be divided into four layers along the direction of coating thickness. The bottom layer is 100 vol.% NiCoCrAlY bond coat. The middle gradient layer corresponds to 20 vol.% NiCoCrAlY/80 vol.% YSZ and 40 vol.% NiCoCrAlY/60 vol.% YSZ, respectively. The ratio of NiCoCrAlY/YSZ is 2:8 and 4:6 in Figure 2a–e.

The top layer is the 100 vol.% YSZ top coat. The thickness of TBCs ranges from 1000 to 3000 μm. In addition, a different ratio of NiCoCrAlY and YSZ for middle gradient layer is also be prepared and shown in Figure 2f–h, the corresponding ratio is 4:6 and 6:4, respectively. The thickness of TBCs ranges from 1500 to 2500 μm. Besides, some vertical cracks for G6–G8 coatings are also found in the YSZ coating, which derived from the release of residual stress.

Figure 3 displays higher magnification images of the layer gradient in the NiCoCrAlY/YSZ coating. As seen from Figure 3a–g, the layer gradient mainly consists of some lamellar structures and pores. These pores are non-uniformly distributed in the lamellar structure of the coatings, which is due to the incomplete stacking of flattened YSZ-/NiCoCrAlY-droplets and entrapment of some gas phase at the lamellar interface. Besides, these pores content increases with the increase of coatings thickness. Compared to the gradient layer of G1–G4, the pores content within the G6–G8 obviously increases.

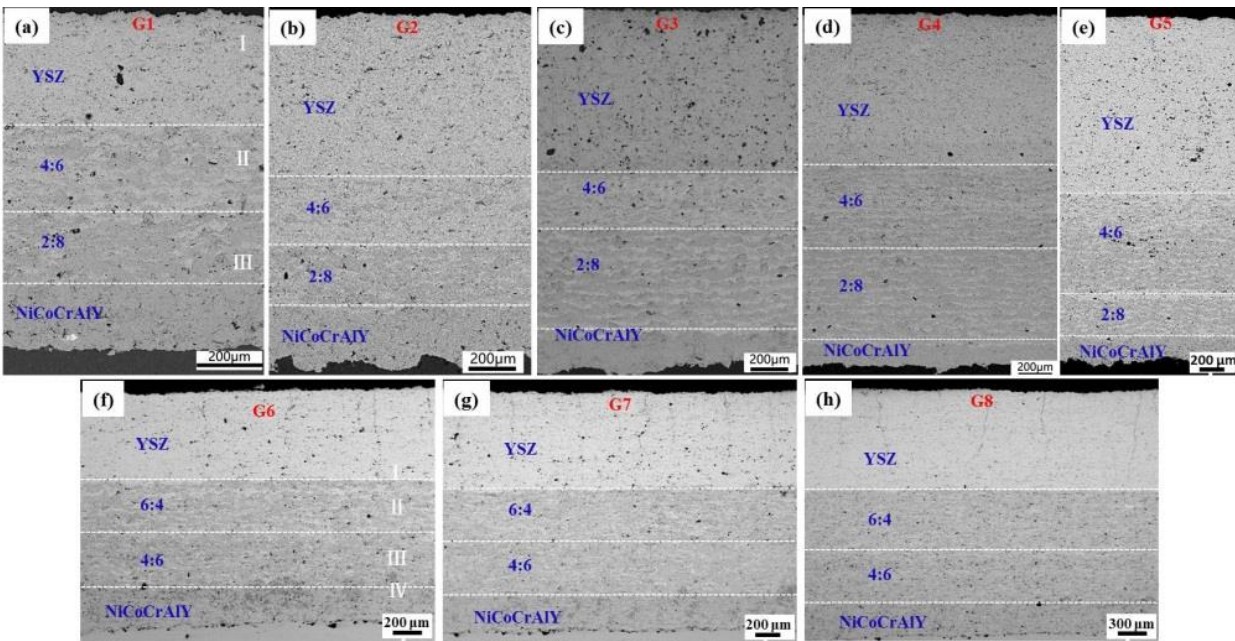

**Figure 2.** Cross-sectional SEM images of gradient NiCoCrAlY/YSZ coating with thickness of (**a**) G1, (**b**) G2, (**c**) G3, (**d**) G4, (**e**) G5, (**f**) G6, (**g**) G7 and (**h**) G8.

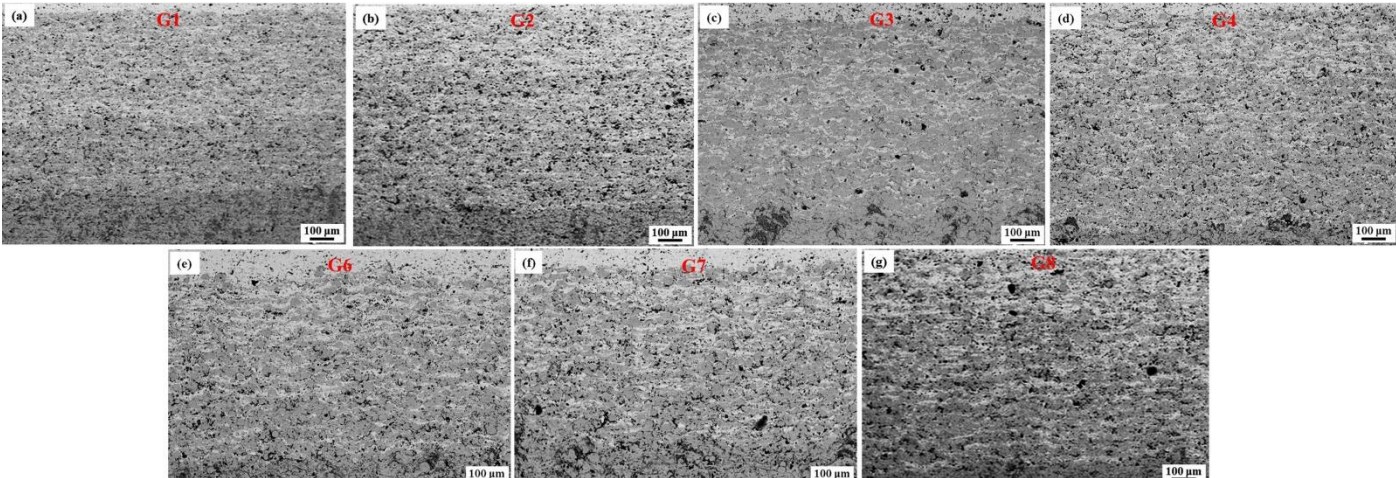

**Figure 3.** Higher magnification images of gradient layer in the NiCoCrAlY/YSZ coating: (**a**) G1, (**b**) G2, (**c**) G3, (**d**) G4, (**e**) G6, (**f**) G7 and (**g**) G8.

Figure 4 gives the pores content of gradient coatings after the measurement by the image analysis (IA) and Archimedean methods. The results from the IA method indicates that the pores content for G1–G5 gradient coatings increases significantly from 4.2% to 4.8%, but in the G6–G8 gradient coatings it declined slightly from 5.9% to 5.7%. The measured value using the Archimedean method is relatively larger than that of the IA method. Comparing the G1–G8 gradient coatings, the change tendency of pores content is similar.

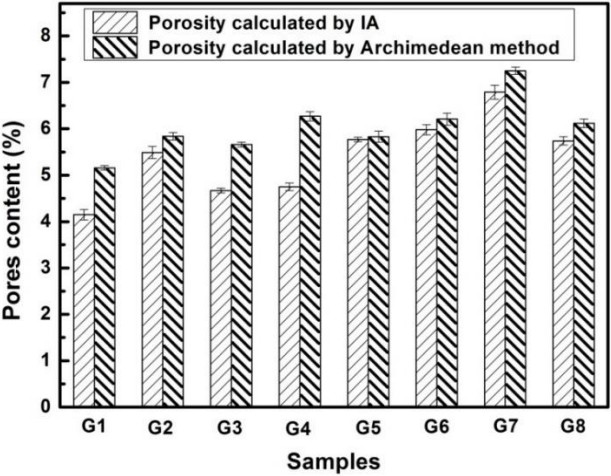

**Figure 4.** Porosity of as-sprayed gradient coatings determined by the image analysis (IA) and Archimedean methods.

Moreover, as seen from Figure 3 and Figure 4, the pores size is smaller and presents a coexisting of micro or submicron multi-scale. Micron-sized pores are attributed to the incomplete wetting behavior of contact interface. Some gas is entrapped in the concave region of a rough surface and this hindered locally the direct contact of flattened YSZ-/NiCoCrAlY- droplets with the substrate. Submicron-sized pores are originated from the air entrapment accompanied by a rapid solidification rate of droplet [20]. These multi-scale pores directly affected the bonding or adhesion of gradient coatings [21].

### 3.2. Bonding Strength

The bonding strengths and fraction location of the samples with millimeter level thickness according to the tensile tests are listed in Table 2. Each sample was measured the three times. The fracture morphologies of coatings are shown in Figure 5.

**Table 2.** Bonding strength and fraction location of samples with millimeter level thickness.

| Samples | Thickness (mm) | Bonding Strength (MPa) | Fraction Location | Average Bonding Strength (MPa) |
|---|---|---|---|---|
| G1-1 | | 44.43 | 100% BC | |
| G1-2 | 1 | 43.35 | BC/GC | 44.54 |
| G1-3 | | 45.84 | BC/GC | |
| G2-1 | | 37.19 | GC,BC/GC | |
| G2-2 | 1.5 | 35.21 | GC,BC/GC | 36.83 |
| G2-3 | | 38.09 | BC/GC | |
| G3-1 | | 31.68 | GC/TC | |
| G3-2 | 2 | 23.97 | GC/TC | 32.59 |
| G3-3 | | 24.10 | GC/TC | |
| G4-1 | | 28.37 | 100% TC | |
| G4-2 | 2.5 | 30.68 | 100% TC | 29.40 |
| G4-3 | | 29.16 | 100% TC | |
| G5-1 | | 20.18 | 100% TC | |
| G5-2 | 3 | 22.20 | 100% TC | 21.10 |
| G5-3 | | 20.92 | 100% TC | |
| G6-1 | | 27.42 | 100% BC | |
| G6-2 | 1.5 | 21.98 | BC/GC | 24.62 |
| G6-3 | | 24.45 | BC/GC | |
| G7-1 | | 24.36 | 100% BC | |
| G7-2 | 2 | 23.21 | GC/TC | 23.50 |
| G7-3 | | 22.95 | GC/TC | |
| G8-1 | | 17.17 | GC/TC | |
| G8-2 | 2.5 | 26.77 | 100% TC | 21.44 |
| G8-3 | | 20.39 | 100% TC | |

As seen from Table 2 and Figure 5a,b, the fracture zone in G1 coating is initially located at the interface of NiCoCrAlY bond coat/substrate and gradually shifts to the YSZ top coat in the G5 coating. For instance, the fracture zone of 1000-μm-thickness coating is located at the bond coat/substrate interface, while the 1500-μm-thick coating fractured along the interface between NiCoCrAlY bond coat and gradient layer. The fracture of 2000-μm-thick coating occurred at the gradient layer/YSZ top coat interface, while one layer of YSZ from 2500-μm-thickness and 3000-μm-thickness coating adheres to the surface of holder, indicating that the fracture zone is located within the YSZ top coat. The above results suggest that the fracture zone gradually shifts from the gradient layer to the YSZ topcoat with increasing the coating thickness. The corresponding bonding strength is gradually reduced. For the G6–G8 coatings, the bonding strength tendency is similar to the above results. The maximum value is 44.5 MPa, which is observed in the 1000-μm-thickness coatings. Furthermore, Figure 5c gives the bonding strength and residual stress as a function of coating thickness. The option of G1–G4 gradient coatings with the higher bonding is regarded as a typical one and can basically reflect the bonding essence. As shown in Figure 5c, as the coating thickness increases from 1000 to 2500 μm, the bonding strength gradually decreases from 44.5 to 29.4 MPa, while the residual stress correspondingly increases from 191 to 287 MPa. Therefore, the residual stress is highly influenced on the bonding strength of coatings.

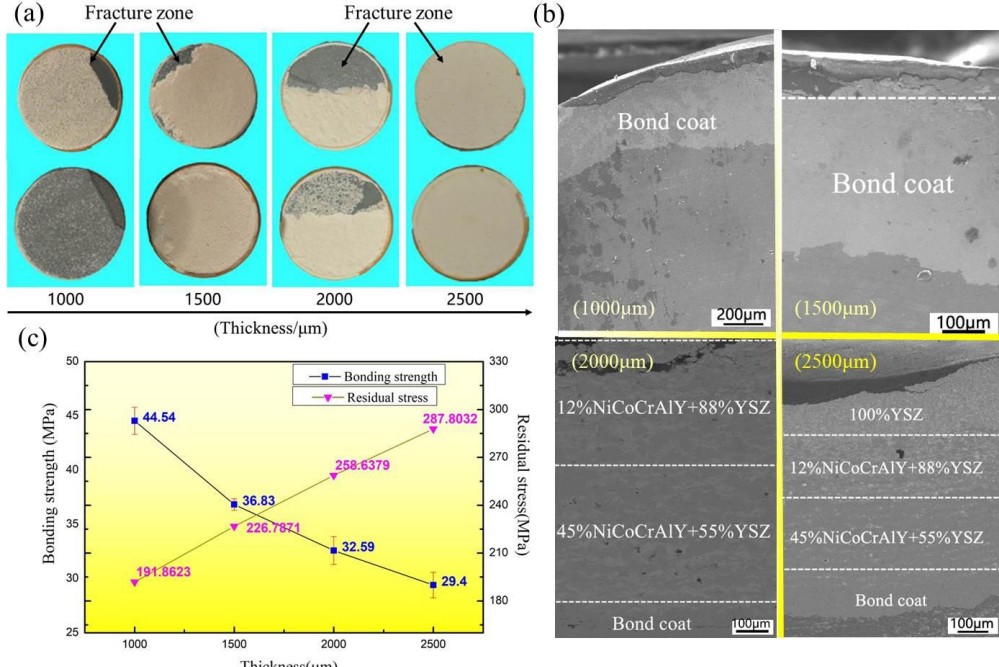

**Figure 5.** (**a**) Fracture morphology of samples after uniaxial tensile test; (**b**) SEM images of polished cross-section in (**a**); (**c**) Bonding strength and residual stress for the coatings with various thicknesses.

The reason for the decrease of bonding strength is closely related to the accumulation of residual stress during coating deposition [22,23]. The residual stress mainly originates from two sources: the first one is quenching stress ($\sigma_{quenching}$) resulting from the fast formation of lamellar structures due to the spreading and solidification of molten particles that have impacted onto the surface of the substrate or the already formed the lamellar structures. The second one is the thermal mismatch stress caused by the thermal expansion coefficient mismatch between coating and substrate, which is also named as the cooling stress ($\sigma_{cooling}$) and mainly affects the intercrystalline bonding of the coating [24,25].

The quenching stress ($\sigma_{quenching}$) and cooling stress ($\sigma_{cooling}$) can be calculated by Equations (2) and (3), respectively:

$$\sigma_{quenching} \approx \alpha_c (T_m - T_s) E_c \tag{2}$$

$$\sigma_{cooling} = \frac{\left[ E_c \left( T_f - T_s \right) (\alpha_c - \alpha_s) \right]}{\left[ 1 + 2 \left( \frac{E_c t_c}{E_s t_s} \right) \right]} \tag{3}$$

where $E_c$ and $E_s$ are the elastic modulus of the coating and the substrate; $\alpha_c$ and $\alpha_s$ are the thermal expansion coefficients of the coating and the substrate; $T_m$, $T_f$ and $T_s$ are the lamella melting temperature, the deposition temperature and the substrate temperature; $t_c$ and $t_s$ are the thicknesses of the coatings and the substrate, respectively. For a nickel-based superalloy/NiCoCrAlY/YSZ system, $T_f$ is much higher than $T_s$ while $\alpha_c$ is smaller than $\alpha_s$. The cooling stress ($\sigma_{cooling}$) is a tensile one while the quenching stress ($\sigma_{quenching}$) is a compressive stresses. The sum of the two results yields a residual stress. As seen from Figure 5c that the residual stress ($\sigma_{residual}$) increases with the increase of coating thickness ($t_s$) accompanied by the decrease of bonding strength. Under the action of tensile stress along the coating thickness direction, many micro-cracks are more easily generated in the stress concentration zone and can rapidly expand along or perpendicular to the coating thickness direction. In other words, the accumulation of residual stress improves the initiation and propagation of micro-cracks, leading to the decrease of bonding strength with the higher thickness coatings after the tensile test.

The residual stress is highly influenced on the microstructure formation and bonding strength of coatings. Generally, the residual stress mainly originates from the quenching stress and the thermal mismatch stress during the coating deposition [7]. The vertical cracks are really a release of residual stress. Some research showed that the vertical cracks through the direction of coating thickness are beneficial to the improvement of bonding strength [8,9]. However, in our work, some vertical cracks are only found in the YSZ top layer of G6–G8 coatings and as seen in Figure 2f–h, the gradient layer and bond layer are not observed. The microstructure results suggested that these vertical cracks can release the residual stress to some extent. In addtion to the vertical cracks, the pores in the coating also influence the bonding strength. For high-thickness gradient NiCoCrAlY/YSZ TBCs with millimeter thickness, some multi-sized pores are found in every layer of the gradient coating. The multi-sized pore content of G6–G8 coatings increases compared to the G1–G5 coatings. These pores will lead to the decrease of bonding strength of G6–G8 although the existence of vertical cracks in the YSZ layer. Therefore, the bonding strength is dependent on the vertical cracks and pores of gradient coatings.

In order to explain the reason for the migration of fracture zone, higher magnification images of gradient NiCoCrAlY/YSZ coating at the different regions (BC, BC/GC, GC, GC/TC and TC) are shown in Figure 6. As seen from Figure 6a, compared to the other gradient coatings (see Figure 6b–g), the structure of the "TC" and "GC" zone for G1 coating is very dense and leads to an improvement of the bonding strength. Moreover, the height of the "GC" layer is much finer and some micro-convex structure of the "GC" layer is found at the "GC/TC" interface and inside the "GC" layer, which is further beneficial to better adhesion with the "TC" layer and results in the fracture location of 100% BC and BC/GC interface of G1 coating. However, in the G2 and G3 coatings, some larger-sized pores appear in the "GC/BC" interface and "GC" zones compared to the G1 coating. These pore structures determine the fracture location of the BC/GC and GC/TC zones and finally the decrease of bonding strength. For the G4 coating, in addition to the above larger-sized pores, some coarse cracks are also found in the "TC" zone. These defects quickly extend and induce a 100% TC fracture in the G4 coatings. Besides, the lower bonding strengths of G6–G8 coatings are basically related to these defects within different zones of the microstructure. The bonding strength of 4:6 and 2:8 ratio of NiCoCrAlY and YSZ layers in the "GC" zone of the G1–G4 is higher than that 6:4 and 4:6 ratio in the G6–G8 coatings, suggesting the gradient ratio in the "GC" layer closely influences the bonding in the supersonic plasma spraying.

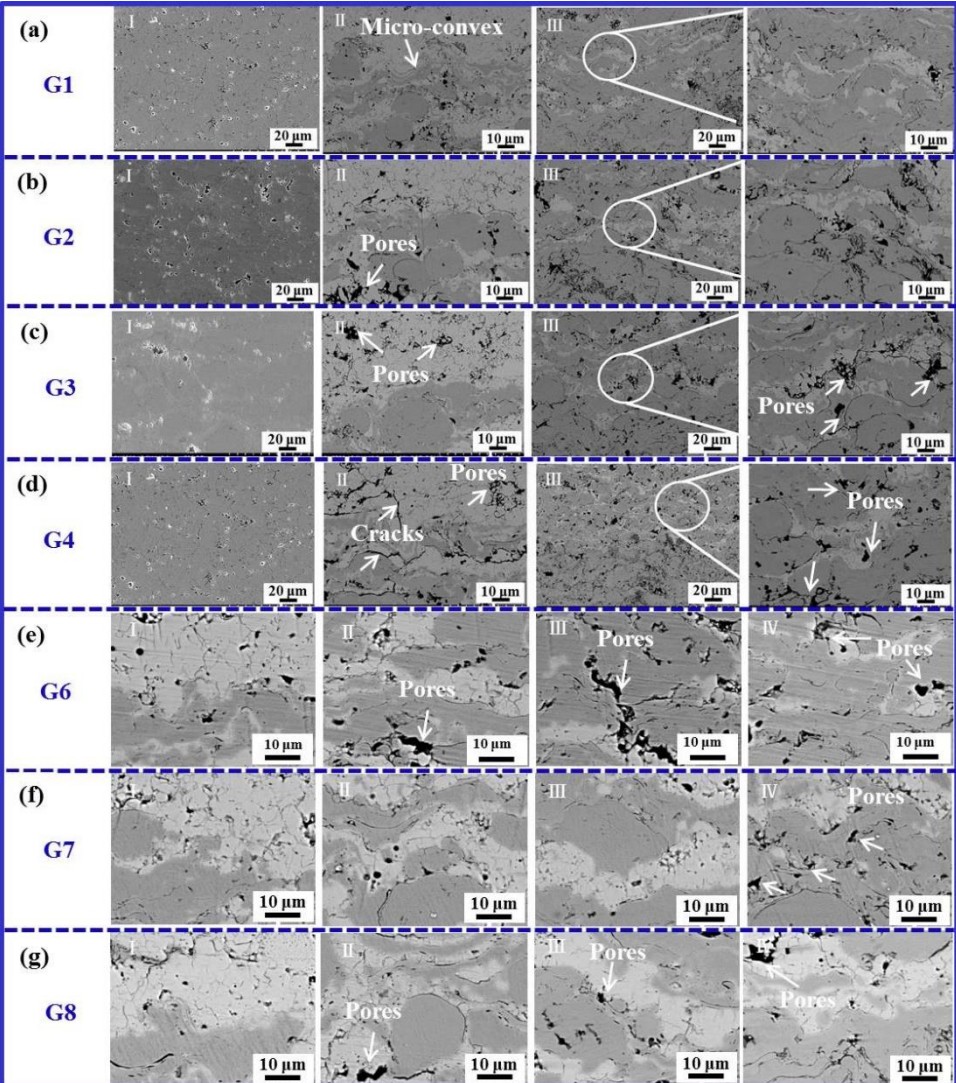

**Figure 6.** Higher magnification images of gradient NiCoCrAlY/YSZ coating at the different regions: (**a**) G1, (**b**) G2, (**c**) G3, (**d**) G4, (**e**) G6, (**f**) G7 and (**g**) G8.

Table 3 lists the bonding strengths of TBCs from different studies. As shown in Table 3, the composition, thickness and spraying method have a significant effect on the bonding strength of TBCs. In addition, it very difficult to reach 45 MPa for the TBCs with thickness above 1000 μm. In this work, the gradient TBCs are deposited by a synchronous dual powder feeding system. In this system, the starting materials NiCoCrAlY and YSZ are simultaneously injected to the different regions of plasma jet, which can guarantee the best melting state of starting materials and avoid any over-melting of NiCoCrAlY and semi-melting of YSZ. Based on this system, the gradient transition of NiCoCrAlY→YSZ and microstructural homogeneity of gradient zone can be tailored, which effectively decreases the residual stress and increases the mechanical bonding of substrate/bond coat/gradient zone/top coat system, leading to a higher bonding strength of SAPS gradient TBCs than other counterparts.

**Table 3.** Bonding strengths of TBCs from different studies.

| Study | Composition | Spraying Method | Bonding Strength/MPa |
|---|---|---|---|
| X. Fang [15] | NiCrCoAlY (50 μm)–YSZ (400 μm) | SPS | 10.83 |
| M.M. Dokur [26] | NiCoCrAlY (100 ± 10 μm)–CYSZ/Al$_2$O$_3$ (12 layers)–YSZ (400 ± 20 μm) | HVOF + APS | 11.5 ± 1.7 |
| K.A. Khor [27] | NiCoCrAlY (150 μm)–75%NiCoCrAlY + 25% YSZ(200 μm)–50% NiCoCrAlY + 50% YSZ (200 μm)–YSZ (200 μm) | APS | 23 ± 2 |
| R.Ghasemi [28] | NiCrAlY (250 μm)–conventional YSZ (250 μm) | APS | 25.25 |
| R.Ghasemi [28] | NiCrAlY (250 μm)-nanostructure YSZ (250 μm) | APS | 38.21 |
| This study | NiCoCrAlY (220 μm)–45% NiCoCrAlY + 55%YSZ (200 μm)–12% NiCoCrAlY + 88%YSZ (250 μm)–YSZ (330 μm) | SAPS | 44.54 |

SPS: supersonic plasma spraying; APS: atmospheric plasma spraying; HVOF: high-velocity oxy-fuel spraying; SAPS: supersonic atmospheric plasma spraying.

### 3.3. Thermal Conductivity

Figure 7 gives the thermal conductivity of G1–G5 gradient coatings at a function of different temperatures. At the temperature of 200–600 °C, the thermal conductivities of G1–G5 gradient coatings are gradually decreased. However, at the temperature of 600–1000 °C, the corresponding thermal conductivities increase except for the G5 coating. This is due to the radiative heat transfer took place through the material during the measurement of thermal conductivity at high temperatures [29,30].

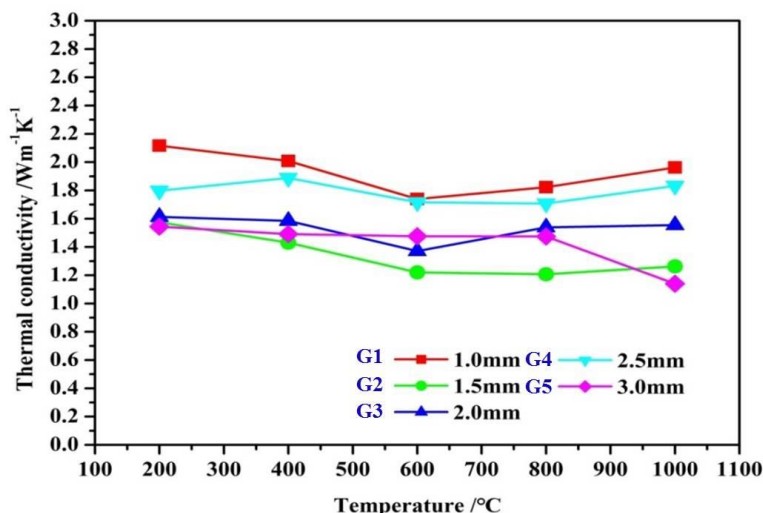

**Figure 7.** Thermal conductivity of gradient coatings at the different temperature.

Furthermore, the G2 gradient coating shows the lowest thermal conductivity, which is mainly attributed to some horizontal micro-cracks that are parallel to the substrate in the region of the "GC" layer (see Figure 4). Although relatively high porosity is also found in the coating, these micro-cracks usually act as the lateral thermal diffusivity and effectively prevent the substrate from heating.

### 3.4. Thermal Cycling Property and Microstructural Evolution

The shock resistance of coatings is evaluated by air-quenching at 1050 ± 20 °C. Figure 8 gives the surface morphology of gradient coatings after different thermal cycle tests. As seen from Figure 8, the thermal cycling life of G1 coating is higher than that of other ones. After 51 thermal cycles, the failure behavior is regard as the separation of coating and substrate. The main reason is that the mismatch of thermal expansion coefficients of both

the coating and substrate [31]. For the G2 and G4 coatings, some obvious net cracks appear on the surface of failed coatings. After 30 thermal cycles, a lot of spalled spots are found at the periphery of G3 coating. The failure location of G2–G4 is similar to that of the G1 coating, which is also occurred the interface of coating and substrate.

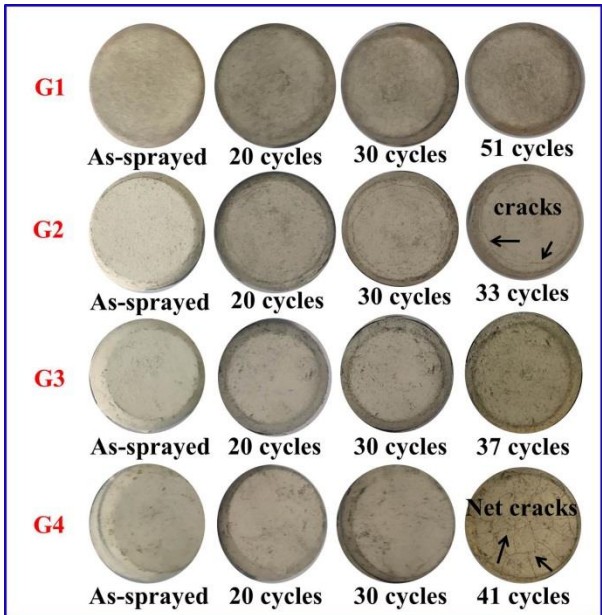

**Figure 8.** Surface morphologies evolution of samples during thermal cycling test.

Figure 9 depicts the fractured morphology of failed coating for different layers after thermal cycle testing. As seen from Figure 9a, the columnar grains in the YSZ layer for G1–G4 coatings have a constricted deformation and grow in both in-plane and through-thickness directions due to the high-temperature sintering effect [32]. The shrinkage of columnar grains within a splat was usually accompanied by grain boundaries grooving [33]. This phenomenon is highly dependent on the diffusion of surface and grains boundary [34,35]. Meanwhile, some coarse cracks are found in the lamellar interface of YSZ layer. The accumulated sintering stress is provided the driving force for the initiation and propagation of cracks [36,37].

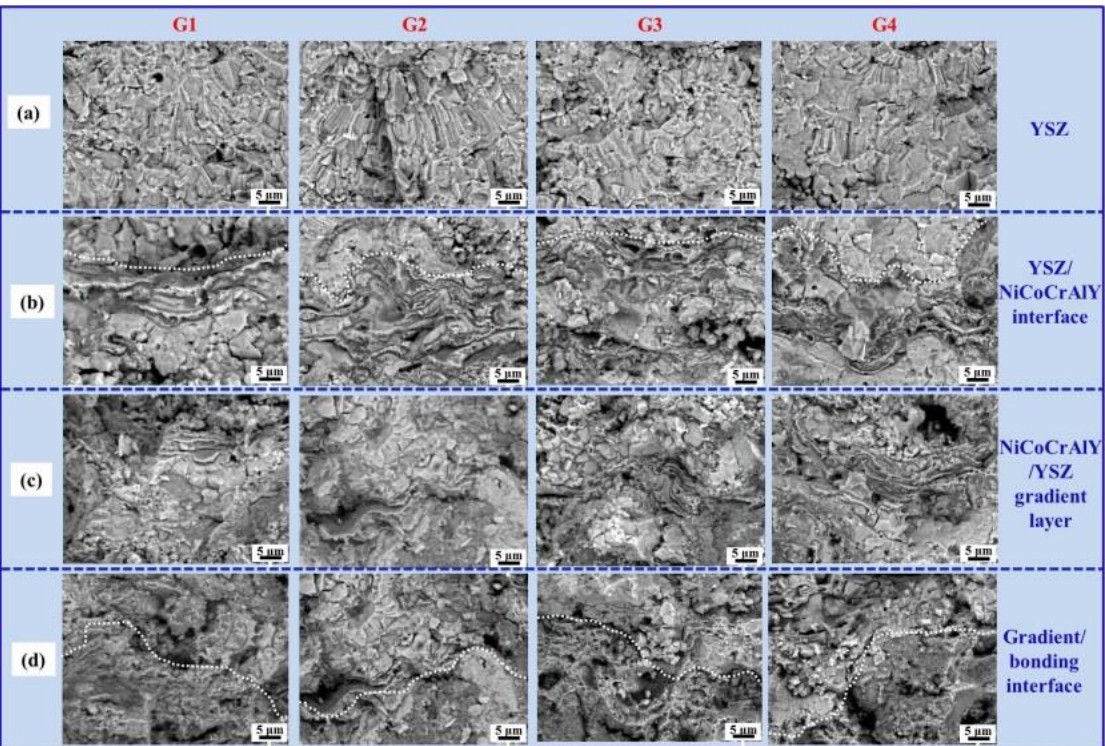

**Figure 9.** Fractured morphology of failure coating for different layers after thermal cycles test: (**a**) YSZ layer; (**b**) YSZ/NiCoCrAlY interface; (**c**) NiCoCrAlY/YSZ gradient layer; (**d**) NiCoCrAlY/YSZ gradient layer and NiCoCrAlY bonding interface.

For the YSZ/NiCoCrAlY interface layer, the bonding of G1 coating is good and the contact zone is relatively smooth (see Figure 9b). Compared to the YSZ/NiCoCrAlY interface at the upper region, as shown in Figure 9c,d, the coarse cracks and porosity in the NiCoCrAlY/YSZ gradient layer and gradient/bonding interface at the bottom region are gradually increased. The higher defect contents of the bottom region is due to the thermal expansion-induced stresses [31]. Generally, the TBCs are subjected to sudden quenching by the cold air, which is produced the transient tensile stresses on the surface of coating and comparative transient compressive stresses in the interior of coating. When these stresses values at the crack tip are exceed the fracture toughness of materials, these cracks will propagate in the destabilized field [38].

During thermal cycling test of gradient coatings, thermal stress ($\sigma_T$) is derived from thermal expansion mismatch between coating and substrate should be considered as the main factor for coating failure, which can be expressed by the following equation [39]:

$$\sigma_T = (\Delta\alpha \times \Delta T \times E_m)/(1 - \nu_m) \tag{4}$$

where $\Delta\alpha$ is the difference of thermal expansion coefficient between the coating and substrate, $\Delta T$ is the temperature difference, $E_m$ and $\nu_m$ are the elastic modulus and poisson ratio of coating, respectively.

For the YSZ/NiCoCrAlY gradient coatings, the temperature difference of the bottom region has a higher value, therefore, the thermal expansion mismatch stress at the bottom region is higher than that of the surface sintering stress at the upper region, which finally leads to the separation of coating and bonding layer. The failure location is consisted with the fracture location after tensile tester, which manifests the higher stress in this region.

Comparing the thermal cyclic life of the above gradient coatings, the G1 coating shows a highest value. This is mainly due to some micro-convex structures at the "GC/TC" interface and inside the "GC" layer, which is more beneficial to the improvement of coating cohesion. The higher cohesion is hardly hierarchically peeled off in the gradient structure

than that of the adhesion between coating and substrate. Therefore, the 1000-μm-thickness gradient NiCoCrAlY/YSZ coating has comprehensive properties. For further research, the high-thickness gradient NiCoCrAlY/YSZ TBCs with the millimeter level should focus on the precise gradient ratio for NiCoCrAlY/YSZ coating through a synchronous dual powder feeding system.

## 4. Conclusions

In the present study, high-thickness gradient NiCoCrAlY/YSZ TBCs with millimeter level are fabricated using a synchronous dual powder feeding system. The main conclusions are as follows:

(1) The 1000-μm-thickness gradient NiCoCrAlY/YSZ TBCs coatings with some micro-convexity showed a higher bonding strength and thermal cycling performance compared to common TBCs.

(2) The residual stress of the coating increased with the increase of coating thickness, which was accompanied by a decrease of bonding strength. The bond strength and residual stress of gradient coatings thus had a negative correlation.

(3) The residual stress at the different layers led to different fracture positions, which gradually shifted from the NiCoCrAlY bond coat to NiCoCrAlY/YSZ the transition zone and finally to the YSZ top coat when the coating thickness increased from 1000 to 3000 μm.

(4) Some coarse cracks initiated and propagated at the bottom region of TBCs, which were originated by thermal expansion mismatch stress and caused the failure of the gradient coating between the "BC" layer and substrate.

These results provide a guideline for the structural design for high-performance coatings.

**Author Contributions:** Drafting the work or revising it critically for important intellectual content, analysis, or interpretation of data for the work, Y.W.; analysis and interpretation of data for the work, Q.L.; data collection, Q.Z.; literature search and date Analysis, T.L.; data collection, N.C.; conception or design of the work, final approval of the version to be published, agreement to be accountable for all aspects of the work in ensuring that questions related to the accuracy or integrity of any part of the work are appropriately investigated and resolved, Y.B. All authors have read and agreed to the published version of the manuscript.

**Funding:** This work was supported by National Science and Technology Major Project (2019-VII-0007-0147), National Natural Science Foundation of China (Grant Nos. 52005388 and U2004169), Collaborative Innovation Center of Advanced Control Valve Project (Grant No. WZYB-XTCX-001), National Key R&D Program of China (Grant No. 2018YFB2004002), China Postdoctoral Science Foundation (Grant No. 2019M653598) and Natural Science Foundation of Shaanxi Province (Grant Nos. 2019TD-020, 2019JQ-586 and 2020JM-631).

**Institutional Review Board Statement:** Not applicable.

**Informed Consent Statement:** Not applicable.

**Data Availability Statement:** The datasets generated during and/or analysed during the current study are available from the corresponding author on reasonable request.

**Conflicts of Interest:** The authors declare no conflict of interest.

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
