# Peer review of "Bonding and Thermal-Mechanical Property of Gradient NiCoCrAlY/YSZ Thermal Barrier Coatings with Millimeter Level Thickness"

_coatings, doi:10.3390/coatings11050600_

Round 1

Reviewer 1 Report

This manuscript from Wang et al. shows their investigation of the bonding and thermal-mechanical properties of millimeter-thick Gradient NiCoCrAlY/YSZ thermal barrier coatings. NiCoCrAlY/YSZ coating is a promising thermal barrier coating for applications in the industry for protecting metallic components from high experimental temperature. In this paper, the authors create the NiCoCrAlY/YSZ coating using the supersonic atmospheric plasma spray technique, and they also showed their study on the boding and thermal-mechanical testing results.

Even though the concepts and the plasma sprayed thermal barrier coating are not new, this manuscript is a complete study that covers the coating method, film density analysis (porosity), bonding strength and residue stress, and thermal conductivity. They also show the surface morphologies evaluation as a function of thermal cycling.  

In summary, I suggest publishing this paper after minor revisions in both figures and texts to improve readability, as shown in the letter to the authors below.

Author Response

Dear Reviewer:

        We have advised the questions by point-by-point response. Please see the attachment.

Reviewer 2 Report

The manuscript presents the study of high-thickness gradient NiCoCrAlY/YSZ TBCs fabricated by a synchronous dual powder feeding system.

The bonding strength, thermal cycling life, as well as residual stress of gradient NiCoCrAlY/YSZ TBCs were analyzed and correlated to the coating thickness. The paper is clearly written, the conclusions are supported by the results.

I would suggest only a minor point regarding the conclusions, not to be presented as in a report form.
The paper should be accepted after this minor correction.

Author Response

Dear Reviewer:

Reviewer 3 Report

This paper investigated that bonding and thermal-mechanical property of gradient  NiCoCrAlY/YSZ thermal barrier coatings with millimeter thickness, and exhibits very interesting and highly scientific value. This is also a highly reliable paper, with bonding strength measurements repeated three times. Therefore, reviewer recommends to accept this paper.

“As shown in Fig. 5c, as the coating 184 thickness increases from 1000 μm to 2500 μm, the bonding strength gradually decreases 185 from 44.5 MPa to 29.4 MPa, while the residual stress correspondingly increases from 191 186 MPa to 287 MPa.”

Figure 2 of (f) G6, (g) G7 and (h) G8 shows that cracks are clearly visible in the vertical direction of the thick YSZ coating, but is this not a release of residual stress?

In addition, these vertical cracks have anything to do with the decrease in the bonding strength?

Reviewer requests to reflect these questions in the paper.

Author Response

Dear Reviewer:

Reviewer 4 Report

1. The literature review should be extended to include other publications in this field. 2. It should be stated whether the results relate to the measurement and calculation of the value of a single sample or a specific number. Are these average values (Figs. 4, 7). If so, please provide the deviation of the values on the charts. 3. The SEM drawings and photos as well as the morphological structure are illegible and the differences in the samples cannot be assessed (Fig. 8). 

Author Response

Dear Reviewer:
